# Space-Flight- and Microgravity-Dependent Alteration of Mast Cell Population and Protease Expression in Digestive Organs of Mongolian Gerbils

**DOI:** 10.3390/ijms241713604

**Published:** 2023-09-02

**Authors:** Dmitrii Atiakshin, Andrey Kostin, Viktoriya Shishkina, Alexandra Burtseva, Anastasia Buravleva, Artem Volodkin, Daniel Elieh-Ali-Komi, Igor Buchwalow, Markus Tiemann

**Affiliations:** 1Research and Educational Resource Center for Immunophenotyping, Digital Spatial Profiling and Ultra-structural Analysis Innovative Technologies, Peoples’ Friendship University of Russia, 6 Miklukho-Maklaya St, 117198 Moscow, Russia; atyakshin-da@rudn.ru (D.A.); andocrey@mail.ru (A.K.); volodkin-av@rudn.ru (A.V.); 2Research Institute of Experimental Biology and Medicine, Burdenko Voronezh State Medical University, 394036 Voronezh, Russia; 4128069@gmail.com (V.S.); earth-mars38@yandex.ru (A.B.); fizvsma@yandex.ru (A.B.); 3Institute of Allergology, Charité–Universitätsmedizin Berlin, Corporate Member of Freie Universität Berlin and Humboldt-Universität zu Berlin, 12203 Berlin, Germany; daniel.elieh-ali-komi@charite.de; 4Fraunhofer Institute for Translational Medicine and Pharmacology ITMP, Allergology and Immunology, 12203 Berlin, Germany; 5Institute for Hematopathology, 22547 Hamburg, Germany; mtiemann@hp-hamburg.de

**Keywords:** space flight, mast cells, tryptase, chymase, Mongolian gerbils, the digestive system

## Abstract

Mast cell (MC)-specific proteases are of particular interest for space biology and medicine due to their biological activity in regulating targets of a specific tissue microenvironment. MC tryptase and chymase obtain the ability to remodel connective tissue through direct and indirect mechanisms. Yet, MC-specific protease expression under space flight conditions has not been adequately investigated. Using immunohistochemical stainings, we analyzed in this study the protease profile of the jejunal, gastric, and hepatic MC populations in three groups of Mongolian gerbils—vivarium control, synchronous experiment, and 12-day orbital flight on the Foton-M3 spacecraft—and in two groups—vivarium control and anti-orthostatic suspension—included in the experiment simulating effects of weightlessness in the ground-based conditions. After a space flight, there was a decreased number of MCs in the studied organs combined with an increased proportion of chymase-positive MCs and MCs with a simultaneous content of tryptase and chymase; the secretion of specific proteases into the extracellular matrix increased. These changes in the expression of proteases were observed both in the mucosal and connective tissue MC subpopulations of the stomach and jejunum. Notably, the relative content of tryptase-positive MCs in the studied organs of the digestive system decreased. Space flight conditions simulated in the synchronous experiment caused no similar significant changes in the protease profile of MC populations. The space flight conditions resulted in an increased chymase expression combined with a decreased total number of protease-positive MCs, apparently due to participating in the processes of extracellular matrix remodeling and regulating the state of the cardiovascular system.

## 1. Introduction

The space industry’s scientific and technical infrastructure is currently improving, enabling an increase in the length of time a human can spend in the circumstances of orbital flight [1]. Today’s understanding of life is largely framed and well studied in terms of how it has evolved and persisted under the gravitational pull of the Earth (1G). Earth’s gravity field becomes inversely weaker with the square of the distance from the planet (also known as the inverse square law), so the pull of gravity in Low Earth Orbit (LEO), where the ISS orbits, is slightly less than on the Earth’s surface [2]. Additionally, astronauts face various health issues during their stay in space largely due to the change in diet, radiation (which may result in genetic mutations and cancer), stress, etc. [3,4]. The ability of space flights to affect human immune cells and, as a result, the orchestration of immune responses, have always been a focus of interest. In line with this, the cytotoxic activity of NK cells obtained from astronauts who spent a long time on the ISS was shown to be reduced notably when compared to normal ground-based counterparts or preflight samples upon being exposed to a panel of tumor cell lines [5]. One of the innate immune cell types is mast cells (MCs), the biology of which we have already comprehensively reviewed [6]. MCs develop from CD34+/CD117+ progenitors which, after being liberated from bone marrow, find their way in circulation according to a finely-tuned chemokine- and integrin-based trafficking to different tissues, including gastrointestinal tissues [7,8,9,10]. In humans, MCs classically are classified according to their content of proteases in two subsets, namely MC_T_ (expressing tryptase) and MC_TC_ (expressing tryptase and chymase) [11]. The data obtained recently using molecular techniques evidence a pronounced polyfunctionality of MC proteases in terms of the state of the extracellular matrix (ECM), target cells, and mechanisms of the tissue microenvironment remodeling [12,13,14,15,16,17,18,19]. Considering the role and significance of MCs in not only allergic events but in non-allergic pathologies, their biology deserves to be studied in microgravity to obtain a better insight into their orchestrated immune network and crosstalk with other components of the immune system. However, to date, there is limited research in this area. The present study aimed to investigate the MC protease profile of the stomach, jejunum, and liver in Mongolian gerbils under 12-day space flight conditions and simulated the physiological effects of weightlessness in ground-based conditions.

## 2. Results

When studying gastric micropreparations of experimental animals, it was found that the animals of the vivarium group were characterized by the prevailing tryptase-positive MC population in the mucous membrane, while sporadic chymase expression was detected. MCs expressing chymase predominated in the MC connective tissue subpopulation localized in the submucosa and muscle membranes. When examining the submucosa, a greater number of chymase-positive MCs were detected; their number was only slightly less than the number of tryptase-positive MCs compared to the level detected in the mucosa (Table 1, Figure 1).

This was also supported by the MC protease profile evidencing a higher content of MCs with the expression of chymase exclusively, as well as MCs with the expression of both proteases simultaneously in the connective tissue of the submucosal, muscular, and serous membranes (Table 2, Figure 1).

A number of MCs were detected in the jejunum of animals from the vivarium group using immunohistochemical tryptase identification. Tryptase-positive MCs formed groups and also happened to be located singly (Figure 2). In the mucosa, they were more often located in the intercryptal stroma. A few MCs were localized at the border of the mucosa and submucosa. In the MC cytoplasm, tryptase-positive granules were well visualized and showed clarification in the central part. There were tryptase-containing granules located at a distance from the MC, this being a reflection of their secretory activity. In addition, different gradations of MC staining during the immunohistochemical tryptase detection came under notice, to a certain extent indicating the different content of this enzyme in MCs. Detection of MC protease profile in the population demonstrated a significant predominance of tryptase-containing cells over chymase-expressing ones. The number of tryptase-positive MCs was prevalent both in the mucosa and in other membranes of the jejunum (Figure 2). Notably, as the analysis of the protease profile of the subpopulation demonstrated, the relative expression of chymase was slightly higher in the lamina propria of the jejunal mucous membrane compared to the MC connective tissue subpopulation.

Further, MCs were predominantly localized in the region of the base of the villi and the muscularis mucosal plate. MCs in the region of the muscular and adventitious membranes were the least likely to express chymase; it was present in single cells.

In the liver of animals, tryptase-positive MCs accounted for an insignificant number, localized mainly within the connective tissue of the portal tracts and central veins, while chymase-positive MCs practically failed to be detected (Table 1, Figure 3).

In the synchronous experiment, the MC protease profile of the stomach and liver walls changed insignificantly. The number of tryptase-positive and chymase-containing MCs differed insignificantly compared to the parameters manifested by Mongolian gerbils from the vivarium control group (Figure 1, 3). After the synchronous experiment, chymase expression increased in MCs located in the submucosa of the jejunum (Table 1, Figure 2). Among the connective tissue MCs, there was a shift in the protease profile towards an increased chymase expression (Table 2).

After the space flight, there was a decreased number of tryptase-positive MCs in the lamina propria of the mucous membrane in the gastric micropreparations (Table 1). However, concurrently, there was a significantly increased number of chymase-positive MCs in the mucosa, both in absolute and relative values, compared to the number of MCs detected in the synchronous experiment (Table 1 and Table 2).

Changes in the protease profile reflected an increased chymase expression after space flight, especially in the mucosa (Figure 1). It should be emphasized that there was an increase in the number of MCs expressing exclusively chymase and in the number of MCs expressing both tryptase and chymase simultaneously (Table 2).

As supported by the analysis of the biomaterial performed after the immunohistochemical tryptase detection, there was a decrease in the population of MCs in the wall of the jejunum. Quantitative calculations demonstrated a decreased volume of tryptase-positive MCs in the jejunal stroma of Mongolian gerbils. This was found in both mucosal and submucosal subpopulations. Tryptase-positive MCs sometimes formed clusters in which they grouped and connected (Figure 2). On the part of tryptase-containing MCs, a pronounced intensification of degranulation was detected (Figure 2). In particular, exocytosis of secretory granules was activated both in the mucous membrane and in other membranes of the jejunum. Concurrently, tryptase-positive granules were found at a sufficient distance from MCs, being freely localized in the intercryptal stroma of the mucosa. Evidently, after the space flight, the effects of intense degranulation could lead to a decreased volume of the MC cytoplasm, reflecting the entry of biologically active substances into the stroma. Occasionally, tryptase-positive granules were only one row in the cytoplasm, located perinuclearly. Tryptase-containing MCs were also detected in the stroma of the villi. After the space flight, there was an increased representation of chymase-containing MCs both in the lamina propria of the mucous membrane and other membranes of the jejunum (Table 1). The number of MCs containing both proteases or exclusively chymase increased significantly compared with the indices of animals from the groups of synchronous experiment and vivarium control, primarily within the submucosal and muscular membranes.

In the livers of animals from the flight group, the representation of chymase-containing MCs increased (Table 1, Figure 3). This was due to both an increased representation of chymase-positive MCs and MCs with simultaneous expression of chymase and tryptase (Table 2). An increased level of chymase production should be considered an important compensatory mechanism for organ adaptation to space flight conditions.

After antiorthostatic suspension, the number of chymase-containing MCs in the gastric mucosa increased compared to the parameters detected in the animals from the vivarium group (Table 1). The protease profile detection demonstrated that chymase expression increased both in the mucosal and regular MC subpopulations (Table 2, Figure 1). This was especially noticeable in the MCs of the mucosal lamina propria. Notably, the absolute amount of tryptase-containing MCs, which were well identified in the stomach wall, practically had no changes compared to the parameters of animals from the vivarium group (Table 1).

Regarding the jejunal mucosa, we can only note a forming trend towards a decrease in the number and some increase in the submucosa and muscular membranes of tryptase-positive MCs. There were often observed patterns of release of tryptase-containing granules (Figure 2). Attention should also be drawn to the increased level of chymase-positive MCs both in the mucosa and submucosa.

As the study of the protease profile of the MC population demonstrated, there was an increased chymase expression in MCs. Interestingly, MCs with chymase expression, as well as MCs with both proteases simultaneously, were detected with greater frequency. Accordingly, the number of MCs expressing tryptase only significantly decreased. We found patterns of exocytosis of chymase-positive granules that were not so pronounced in animals from the vivarium control group.

In the liver, the conditions of antiorthostatic suspension resulted in the activation of chymase expression, which manifested itself both in chymase-containing MCs and by a significantly increasing number of MCs with simultaneous expression of tryptase and chymase (Figure 3). This coincided with the increased total amount of MCs in the liver, primarily in the region of portal triads, which were detected by histochemical staining with toluidine blue and determining the chloroacetate esterase activity (Figure 3). However, the chymase expression in MCs increased more significantly after the space flight compared to the findings obtained in the simulated experiment (Table 1).

## 3. Discussion

The decrease in the total amount of MCs in the examined organs of the digestive system of Mongolian gerbils after space flight may be associated with several factors, including apoptosis, migration, and increased secretion of specific proteases. In particular, in an experiment with ground-based modeling of space flight factors using a rotary cell culture system, it was shown that microgravity negatively regulates the survival and effector function of MCs [20]. However, the depletion of intracellular reserves of specific proteases should also be considered, which can make it difficult to detect MCs during immunohistochemical staining.

As is known, MCs contain the largest range of peptidases both quantitatively and qualitatively if compared with other cells of the immune system [19,21,22]. MC proteases are localized in the cytoplasm, mainly in granules that are packaged by the serglycin–glycosaminoglycan complex [14,15,16,17,23,24].

Chymase has the potential to convert angiotensin I to angiotensin II, regardless of the angiotensin-converting enzyme, which, of course, is of particular significance for blood pressure regulation [25]. Angiotensin II increases the expression of vascular endothelial growth factor (VEGF) by various cells. In space flight, increased chymase expression can represent a certain compensation for a decrease in tryptase biogenesis, maintaining the hypersecretory state of the stomach in microgravity. Chymase can hydrolytically inactivate peptides such as bradykinin [26], as well as some neuropeptides, including substance P (SP) and vasoactive intestinal peptide (VIP) [27]. This property allows chymase, unlike tryptase, to inactivate certain bronchoconstrictors that may limit MC degranulation. Chymase is involved in inflammation modulation by affecting the activation of a number of interleukins [28]. It also has the potential to cleave allergens and specific blood proteins [29], hepatocyte growth factor [30], and endogenous immune proteins, including interleukin-6 and interleukin-13 [31]. In terms of tissue remodeling, chymase effects are crucial in relation to the local activation of matrix metalloproteinases (MMPs) 1 and 3, followed by degradation of several elements of the extracellular matrix [32]. The presence of heparin resulted in a significant increase in the rate at which chymase activated some MMPs, both by accelerating the cleavage of their precursors and by preventing their further degradation. In particular, chymase may activate MMP-1 directly by cleaving leucine and threonine without any intermediates [33].

The detection of chymase-positive MCs in the digestive organs of Mongolian gerbils significantly expanded the interpretation of their potential activity in the organ in response to external exposure. Even though their content in vivarium animals was insignificant, there was a considerable increase in chymase expression after space flight and, to a lesser extent, after the simulated physiological effects of weightlessness, this fact attracting attention. This supported the critical role of MC-produced chymase in adaption to space flight factors, in particular weightlessness.

Thus, increase of chymase in MC leads to a higher number of chymase+ MCs and chymase+/tryptase+ MCs. The detected changes in the MC population are most likely due to changes in gene expression in mast cells of the gastrointestinal tract. Based on earlier studies, we suggest a high degree of plasticity in the MC protease phenotype. In response to external stimuli, MCs can significantly change the biogenesis of preformed secretome components. Obviously, under space flight conditions, the increased need for the physiological effects of chymase leads to the activation of chymase gene transcription processes in a certain pool of MCs localized in the digestive system, which is manifested by an increase in the Tryptase+/Chymase+ phenotype. In addition, an increase in the number of MCs without tryptase expression (with the Tryptase-Chymase+ phenotype) suggests that migrating MC precursors from the bone marrow into the digestive system already have predominant chymase expression.

Tryptase has a wider range of bioeffects that affect the formation of the cellular and extracellular phenotype of an organ-specific tissue microenvironment under certain environmental conditions [12,34].

This largely is due to the expression of its main receptor protease-activated receptor-2 (PAR-2) on different immune and non-immune cells (including endothelial cells [35], spermatozoa [36], and tumor cells [18]). The experimental data obtained over the past few years on the role of tryptase and chymase allow for considering these proteases as promising objects of study in laboratory experiments and in search for new targets for the pharmacological correction of various diseases [13,37,38,39].

The effects of tryptase secretion can be either local, limited to the effect on nearby cells and structures, or more widespread, involving specific organs (for example, in asthmatic bronchospasm), up to generalization, as in systemic mastocytosis or anaphylaxis [40]. It has been found that the intrinsic products of MC biosynthesis can lead to further enhancement of their degranulation by a positive feedback mechanism, potentiating the release of mediators. As demonstrated, an autocrine mechanism of the tryptase-activating effect on mast cell degranulation can also extend to eosinophilic granulocytes [40]. It may have been this particular mechanism that led to significant consumption of mast cell tryptase during the first period of laboratory animals’ stay in weightlessness; this resulted in the depletion of physiological resources, which was observed after space flight.

The cellular and tissue effects of tryptase can be classified as pro-inflammatory and anti-inflammatory [41]. However, in most cases, tryptase is the initiator of inflammatory reactions accompanied by increased capillary permeability and intensive granulocyte recruitment [42]. Tryptase is involved in angiogenesis, degradation of fibrous and amorphous components of the extracellular matrix of connective tissue, and release of growth factors, which can be the results of tryptase-mediated activation of MMPs, mainly MMP-2 and MMP-9 [18]. These endopeptidases play a crucial role in the degradation of structural elements of ECM such as collagen [43,44].

Given the aforementioned aspects of the importance of MC proteases and the fact that cancer and metastasis are among the previously researched effects of prolonged human space flights [45,46], MCs may be involved through tissue accumulation, producing high levels of proteases, leading to dysregulation of MMP production, which ultimately results in excessive damage to the ECM, the liberation of tumor cells, and the progression of metastasis.

Several mechanisms of tryptase effect promoting the growth and differentiation of new blood vessels are discussed in the literature [40,47]. It is likely that the intense secretion of tryptase during the first period of orbital flight leads to additional vessel formation in the gastric and intestinal mucous membranes. This fact, among others, can explain developing mucosal edema, and may also be one of the reasons for implementing the hemodynamic mechanism of the stomach hypersecretory state in microgravity [48]. Notably, generalization of this effect of tryptase may also be provided, since this protease leads to a potentiation of histamine release. Histamine, in turn, stimulates a further increase in tryptase secretion, promoting signal propagation from one group of activated MCs to others [49]. Tryptase also inactivates procoagulant proteins, particularly kininogen and fibrinogen [50]. The increased vascular permeability under the tryptase effect occurs indirectly by increasing the kinin formation. In addition, tryptase induces the expression of interleukin-1 and interleukin-8 from endothelial cells, which can be combined with an increased synthesis of the Intercellular adhesion molecule 1 (ICAM-1) and an increased leukocyte tissue infiltration, as well as the further development of tissue edema in a specific area [40]. This effect is also prolonged due to tryptase’s ability to destroy neuropeptides, including VIP.

Notably, tryptase deficiency in the wall of the stomach and jejunum, developing by the 12th day of space flight, can also result in a deficiency in the mitogenic effect of tryptase on smooth myocytes. This fact, further, may be the cause of thinning of the smooth muscle layer in the wall of the hollow digestive organs and affects the digestive tube motor activity [51].

## 4. Materials and Methods

### 4.1. Experimental Design

The units of analysis were mast cells of the stomach, jejunum, and liver of male Mongolian gerbils (Meriones unguiculatus) divided into several groups. The flight experiment group, implemented within the Foton-M3 12-day orbital flight research project, consisted of 12 animals. The group of the ground-based synchronous experiment included 11 Mongolian gerbils, which stayed for 12 days in the prototype of the flight equipment “Kontur-L” to simulate certain characteristics of space flight circumstances. The third group included 12 vivarium animals. The animals were sacrificed 21 h after landing. Specific physiological effects of weightlessness were simulated using antiorthostatic suspension according to the Ilyin–Novikov technique interpreted by Moray-Holton [52]. This experiment included two groups of animals: of these, 8 animals were subjected to 12-day antiorthostatic suspension, and another 8 animals served as a vivarium group. All stomach, jejunum, and liver tissues were obtained from the Institute of Medical and Biological Problems (IMBP), the Russian Academy of Sciences (Moscow), in compliance with all the requirements for the human treatment of animals, in accordance with the decision of the Commission on Biomedical Ethics of the IBMP (Protocol No. 206 dated 07.10.2007).

### 4.2. Histoprocessing

Fragments of the fundic zone of the stomach, the middle of the left lobe of the liver, and the jejunum were embedded in paraffin according to the standard sample preparation procedure after fixing them for 24–48 h in 10% neutral formalin at room temperature [53,54]. Histological sections with 2 μm thickness were prepared for immunomorphological analysis using an Accu-Cut^®^ SRM™ 200 semi-automatic rotary microtome (Sakura Finetek Europe B.V., Japan). In the stomach and jejunum, we performed a separate study of the mucosal and connective tissue subpopulations of MCs localized in the mucosa and the interstitium of other membranes, respectively. In the liver, MCs were represented only by a connective tissue subpopulation.

### 4.3. Tissue Probe Staining

Immunohistochemical staining was used to identify MC proteases. Single and multiple immunofluorescence labeling was performed according to standard protocols [53]. Concurrently, according to the standard immunolabeling protocol [53], tryptase was detected using mouse monoclonal antibodies (Table 3). Homologous mouse immunoglobulins were blocked during pre-incubation of sections with unconjugated Fab fragments (goat anti-mouse IgG, Jackson ImmunoResearh, #115–007-003, 1:13, Table 4). Rabbit polyclonal antibodies conjugated with biotin were used to stain chymase MCs (Table 3). The list of secondary antibodies and other reagents used in this study is presented in Table 4. A kit with 3.3’-diaminobenzidine as a substrate was used to detect secondary antibodies conjugated with HRP (Table 4). Before the preparation embedding, the nuclei were stained with Mayer’s hematoxylin.

Protease colocalization in MCs was determined by fluorescence microscopy after incubation with primary antibodies diluted 1:500 to chymase and 1:2000 to tryptase overnight at +4 °C. Then, using streptavidin (Streptavidin AlexaFluor 488, # S11223), biotin-bound anti-chymase antibodies were identified and visualized using an appropriate filter. Cy3-conjugated secondary antibodies (Cy^TM^3-conjugated AffinPure Goat Anti Mouse IgG, #115–165-166, 10 mg/mL PBS) were used to detect anti-tryptase primary antibodies. The nuclei were counterstained with DAPI and the sections were mounted in Vectashield temporary mounting medium (Table 4). The staining for fluorescence microscopy was performed according to the standard protocol technique [53].

### 4.4. Image Acquisition

Stained tissues were studied using a hardware–software complex based on a ZEISS Axio Imager.A2 research microscope (Carl Zeiss Microscopy, Jena, Germany) with a documentation system. The ZEN 2.3 software (blue edition, Carl Zeiss, Jena, Germany) was used to analyze the obtained images using a monochrome Camera Axiocam 503 mono (for fluorescence microscopy) and a Camera Axiocam 506 color (for bright-field microscopy) (Carl Zeiss Microscopy, Jena, Germany). MCs were calculated in fields of vision measuring 700 × 500 μm, obtained using a 20x lens. The studied areas had no intersections with each other on the micropreparations. A minimum of 15 fields of vision were investigated in each section. When calculating the parameters of MC population features, both absolute and relative values were used, calculated as a percentage of the total number. 

### 4.5. Statistical Analysis

Statistical analysis was performed using the SPSS software package (V. 13.0, IBM, New York, NY, USA). The results are presented as mean (M) ± m (standard error of the mean). To assess the significance of the differences between the two groups, Student’s *t*-test or Mann–Whitney U test in the case of a nonparametric distribution was used.

## 5. Conclusions

Thus, the high content of tryptase and chymase in MCs provides their active participation in coordinating the integrative–metabolic state of the connective tissue, including microgravity during space flight [55]. After the orbital flight and the ground-based simulated physiological effects of weightlessness, the ratio of proteases in mast cells of the digestive system of Mongolian gerbils changed due to increased chymase expression. This evidences the close involvement of MCs located in the organs of the gastrointestinal tract in the processes of adaptation to space flight factors, including the immune and stromal landscapes of the specific tissue microenvironment of the studied digestive organs. Modification of the MC protease profile has a direct effect on the adaptive remodeling of the extracellular matrix of connective tissue under altered gravity.

## Figures and Tables

**Figure 1 ijms-24-13604-f001:**
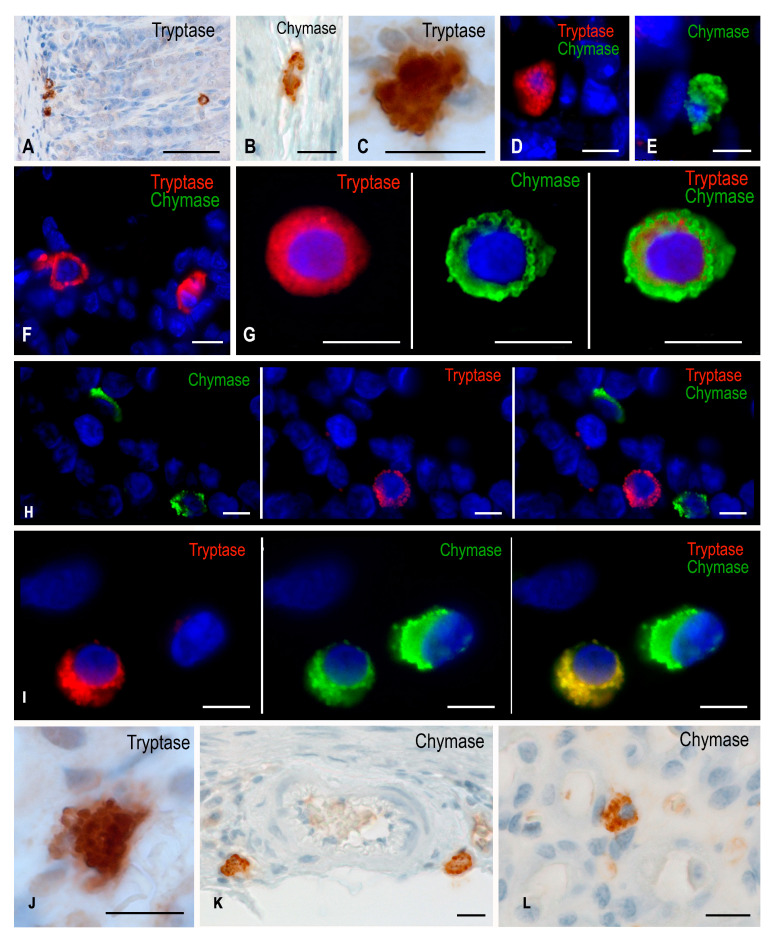
Gastric mast cells of Mongolian gerbils. Experimental groups: (**A**–**E**) vivarium control, (**F**,**G**) synchronous experiment, (**H**,**I**) space flight, (**J**–**L**) antiorthostatic suspension. Notes: (**A**) Localization of tryptase-positive MCs at different levels in the lamina propria. (**B**,**C**) Chymase-positive (**B**) and tryptase-positive (**C**) MCs in the submucosa. (**D**) Attachment of MCs to the gastric proper glands of the connective tissue subpopulation; immunohistochemical staining of mast cell tryptase. (**E**) Secretion of chymase to submucosal fibroblast (presumably). (**F**) Localization of tryptase-positive mast cells in the gastric mucosa. (**G**) A submucosal mast cell containing both tryptase and chymase, distributed predominantly along the periphery of the secretory granules. (**H**) The appearance of chymase-positive MCs in the area of the proper glands of the gastric mucosa. (**I**) Two distinct MC phenotypes in the submucosa, with visualized chymase secretion into the extracellular matrix. (**J**) A mucosal mast cell filled with large tryptase-positive granules, with evident secretion into the intercellular substance. (**K**,**L**) Chymase-positive MCs in the perivascular region of the submucosal muscular artery (**K**) and mucosal intrinsic plasty (**L**). Scale: (**A**) 50 µm, the rest—10 µm.

**Figure 2 ijms-24-13604-f002:**
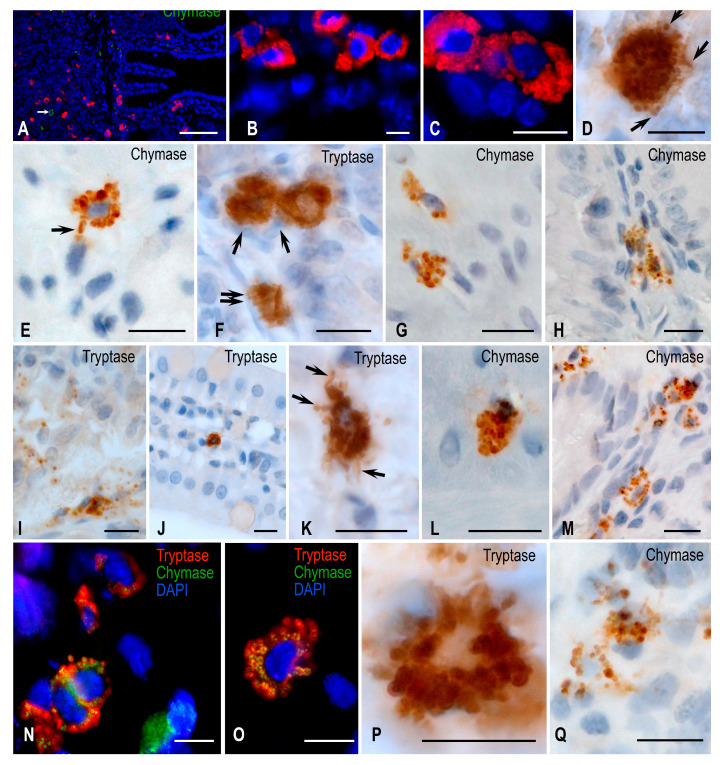
Jejunal mast cells of Mongolian gerbils. Experimental groups: (**A**–**E**) vivarium control, (**F**–**H**) synchronous experiment, (**I**–**O**) space flight, (**P**,**Q**) antiorthostatic suspension. Notes: (**A**) Distribution of tryptase-positive MCs in all structures of the jejunum; chymase-positive MCs are absent in the mucosa. (**B**) Grouping of tryptase-positive MCs. (**C**) Distribution of tryptase along the periphery of contacting mast cell granules. (**D**) A mast cell is filled with tryptase-positive granules. Part of them are transported to the extracellular matrix (arrowed). (**E**) Localization of chymase in large submucosal MC granules; part of them are secreted (arrowed). (**F**) Mucosal mast cells in contact with immunocompetent cells (arrowed) and smooth myocytes of the muscularis lamina (double-arrowed). (**G**) Chymase-positive cells in the muscle layer. (**H**) Chymase-positive MCs at the base of the mucosal villus with signs of degranulation. (**I**) Severe degranulation of tryptase-positive MC granules at the mucosal–submucosal interface. (**J**) Migration of tryptase-positive MCs in the stroma of the jejunal villus. (**K**) Different mechanisms of tryptase degranulation into the extracellular matrix (arrowed). (**L**) Chymase-positive MCs in the muscle layer. (**M**) Increase in the number of chymase-positive MCs in the mucosa. (**N**,**O**) MCs with simultaneous tryptase and chymase expression. (**P**) Tryptase-positive mucosal MCs with signs of degranulation. (**Q**) Active secretion of chymase-positive granules into a specific mucosal tissue microenvironment. Scale: (**A**) 100 µm, the rest—10 µm.

**Figure 3 ijms-24-13604-f003:**
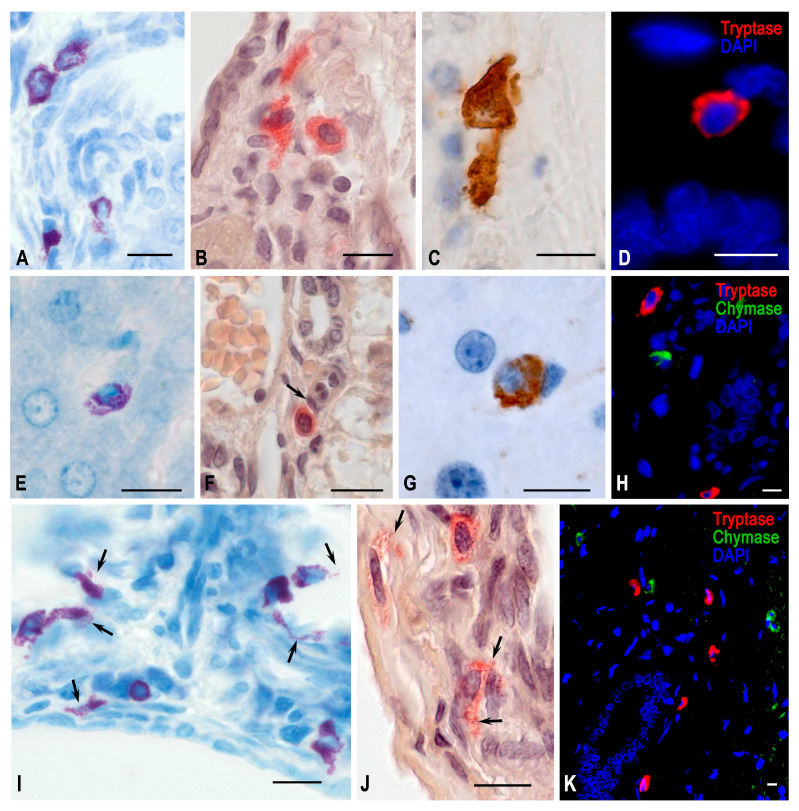
Hepatic mast cells of Mongolian gerbils. Experimental groups: (**A**–**D**) vivarium control, (**E**–**H**) space flight, (**I**–**K**) antiorthostatic suspension. Techniques: (**A**,**E**,**I**) staining with toluidine blue, (**B**,**F**,**J**) determination of chloroacetate esterase activity, (**C**,**D**,**G**,**H**,**K**) immunohistochemical staining of specific MC proteases. Notes: (**A**–**D**) Mast cells in various morphofunctional states in the stroma of portal triads. The contact formation of MCs with each other is typical. (**E**) Degranulation of MCs in the liver lobule. (**F**). MCs in the stroma of the portal tract; close contact with pericyte (arrowed). (**G**) A chymase-positive MC in the liver lobule. (**H**) Chymase- and tryptase-positive MCs in the region of the portal tract. (**I**) High content of mast cells in the periportal region; active secretion of biosynthetic products into the extracellular matrix (arrowed). (**J**) MCs of the portal tract in a state of active secretion (arrowed). (**K**) Mast cells with diverse protease expression in the portal tract region. Scale: 10 µm.

**Table 1 ijms-24-13604-t001:** The content of tryptase- and chymase-positive MCs in the digestive organs of Mongolian gerbils (detected by immunohistochemical staining, per a field of vision).

Groups of Animals	MC Subpopulation	MC Identification Technique					
		Tryptase MCs			Chymase MCs		
		Jejunum	Stomach	Liver	Jejunum	Stomach	Liver
Experiment “Rodentsiya”							
VC	MSMC	39.6 ± 2.2	4.4 ± 0.3	3.2 ± 0.4	2.6 ± 0.4	Single	Single
	CSMC	2.5 ± 0.2	1.5 ± 0.2		Single	1.1 ± 0.2	
SE	MSMC	42.4 ± 2.8	4.3 ± 0.3	2.9 ± 0.2	2.8 ± 0.2	Single	Single
	CSMC	2.8 ± 0.3	1.4 ± 0.1		0.4 ± 0.0 *	1.3 ± 0.1	
SF	MSMC	28.6 ± 1.1 *^,^**	2.6 ± 0.2 *^,^**	2.2 ± 0.2 *^,^**	7.8 ± 0.8 *^,^**	1.2 ± 0.1 *^,^**	1.8 ± 0.2 *^,^**
	CSMC	2.1 ± 0.2 *^,^**	1.4 ± 0.2		2.2 ± 0.1 *^,^**	1.5 ± 0.1 *^,^**	
Simulated physiological effects of 12-day weightlessness							
VC	MSMC	36.8 ± 2.7	4.1 ± 0.7	2.9 ± 0.3	2.9 ± 0.2	Single	Single
	CSMC	2.4 ± 0.3	1.4 ± 0.1		Single	1.2 ± 0.1	
AnOrtSusp	MSMC	33.4 ± 2.5	4.8 ± 0.4	3.8 ± 0.2 *	3.6 ± 0.1 *	0.3 ± 0.1 *	0.8 ± 0.2 *
	CSMC	2.9 ± 0.3	1.7 ± 0.2		0.5 ± 0.1 *	1.4 ± 0.1	

Notes and abbreviations: VC—vivarium control; SE—synchronous experiment; SF—space flight; AnOrtSusp—antiorthostatic suspension; MSMC—mucosal subpopulation of mast cells; CSMC—connective tissue subpopulation of mast cells; *—*p* < 0.05 compared to vivarium control, **—*p* < 0.05 compared to synchronous experiment.

**Table 2 ijms-24-13604-t002:** Protease profile of mast cells in the digestive organs of Mongolian gerbils (detected by multiple immunolabeling, fluorescence microscopy, %).

Groups of Animals	Organs	MC Mucosal Subpopulation			MC Connective Tissue Subpopulation		
		MC_TRY_	MC_CHYM_	MC_TRY+CHYM_	MC_TRY_	MC_CHYM_	MC_TRY+CHYM_
Experiment “Rodentsiya”							
VC	Jejunum	89.8 ± 3.1	6.1 ± 0.2	4.1 ± 0.2	94.3 ± 3.3	3.8 ± 0.2	1.9 ± 0.1
	Stomach	91.1 ± 2.3	3.2 ± 0.2	5.7 ± 0.4	45.4 ± 2.8	27.3 ± 3.4	27.3 ± 3.4
	Liver ^1^	*-*	*-*	*-*	97.2 ± 2.4	0.9 ± 0.2	1.9 ± 0.1
SE	Jejunum	90.6 ± 4.4	6.0 ± 0.4	3.4 ± 0.3	84.8 ± 4.2 *	12.1 ± 1.1 *	3.1 ± 0.2 *
	Stomach	89.4 ± 4.2	3.8 ± 0.3	6.8 ± 0.3	43.4 ± 4.2	26.3 ± 2.3	30.3 ± 2.5
	Liver ^1^	-	-	-	97.1 ± 3.3	0.7 ± 0.1	2.2 ± 0.3
SF	Jejunum	65.7 ± 4.5 *^,^**	17.9 ± 1.1 *^,^**	16.4 ± 1.2 *^,^**	33.3 ± 2.8 *^,^**	34.9 ± 3.4 *^,^**	31.8 ± 2.5 *^,^**
	Stomach	25.4 ± 0.2 *^,^**	28.4 ± 1.9 *^,^**	46.2 ± 3.3 *^,^**	24.2 ± 2.2 *^,^**	36.7 ± 3.6 *^,^**	39.1 ± 3.9 *^,^**
	Liver ^1^	-	-	-	21.3 ± 2.8 *^,^**	22.4 ± 1.2 *^,^**	56.3 ± 1.8 *^,^**
Simulated physiological effects of 12-day weightlessness							
VC	Jejunum	90.1 ± 3.1	5.4 ± 0.3	4.5 ± 0.5	93.4 ± 3.2	4.2 ± 0.3	2.4 ± 0.2
	Stomach	87.6 ± 4.2	6.2 ± 0.5	6.2 ± 0.4	43.2 ± 3.1	25.7 ± 2.7	31.1 ± 1.4
	Liver ^1^	-	-	-	96.5 ± 2.9	1.5 ± 0.1	2.0 ± 0.3
AnOrtSusp	Jejunum	79.5 ± 3.2 *	8.3 ± 0.4 *	12.2 ± 0.9 *	81.4 ± 4.5 *	9.2 ± 0.6 *	9.4 ± 0.5 *
	Stomach	65.0 ± 3.5 *	12.6 ± 1.1 *	22.4 ± 2.0 *	26.2 ± 1.9 *	35.6 ± 3.1 *	38.2 ± 2.2 *
	Liver ^1^	-	-	-	76.3 ± 3.8 *	5.4 ± 0.6 *	18.3 ± 1.1 *

Notes and abbreviation: VC—vivarium control; SE—synchronous experiment; SF—space flight; AnOrtSusp—antiorthostatic suspension; MC_TRY_—tryptase-positive mast cells; MC_CHYM_—chymase-positive mast cells; MC_TRY+CHYM_—mast cells with simultaneous tryptase and chymase expression; *—*p* < 0.05 compared to vivarium control, **—*p* < 0.05 compared to synchronous experiment. ^1^—Hepatic MCs are represented by a connective tissue subpopulation.

**Table 3 ijms-24-13604-t003:** Primary antibodies used in the study.

Antibodies	Host	Catalogue Nr.	Dilution	Source
Tryptase	Mouse monoclonal [AA1] Ab	#ab2378	1:2000	AbCam, Cambridge, UK
Mast cell Chymase/CHYMASE Antibody, Biotin Conjugated	Rabbit Polyclonal Ab	#bs-2353R-Biotin	1:500	Bioss, Woburn, MA, USA

**Table 4 ijms-24-13604-t004:** Secondary antibodies and other reagents.

Antibodies and Other Reagents	Source	Dilution	Label
Goat anti-mouse IgG (#115–007-003)	Jackson ImmunoResearh, West Grove, PA, USA	1:13	w/o
Cy3-conjugated AffinPure Goat Anti Mouse IgG (#115–165-166)	Jackson ImmunoResearh, West Grove, PA, USA	10 Mг/1 Mл PBS	Cy3
Streptavidin (# S11223)	Invitrogen™, Darmstadt Germany	2 mg/mL solution	Alexa Fluor 488
AmpliStain™ anti-Mouse 1-Step HRP (#AS-M1-HRP)	SDT GmbH, Baesweiler, Germany	ready to use	HRP
AmpliStain™ anti-Rabbit 1-Step HRP (#AS-R1-HRP)	SDT GmbH, Baesweiler, Germany	ready to use	HRP
4′,6-diamidino-2-phenylindole (DAPI, #D9542–5MG)	Sigma, Hamburg, Germany	5 µg/mL	w/o
VECTASHIELD^®^ Mounting Medium (#H-1000)	Vector Laboratories, Burlingame, CA, USA	ready to use	w/o
DAB Peroxidase Substrat Kit (#SK-4100)	Vector Laboratories, Burlingame, CA, USA	ready to use	DAB
Mayer’s hematoxylin (#MHS128)	Sigma-Aldrich	ready to use	w/o

## Data Availability

Study data are available from the corresponding author upon reasonable request.

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
