# Peer review of "Space-Flight- and Microgravity-Dependent Alteration of Mast Cell Population and Protease Expression in Digestive Organs of Mongolian Gerbils"

_ijms, 2023, doi:10.3390/ijms241713604_

Round 1
Reviewer 1 Report
Atiakshin et al. studied the mast cell protease expression in digestive organs of mongolian gerbils after space flight conditions. Using immunohistology staining for tryptase and chymase, they showed that after space flight mast cell number decreased in digestive organs. They observed an increase of chymase expression in MCs (including tryptase+/chymase+ MCs and chymase+ MCs) with a decrease of tryptase expression in MC (tryptase+MCs). I would have some points to address.
Major comments:
1- Line 166-169: the authors showed a decrease of tryptase in MCs with some de granulation. It is possible that the tryptase decrease is due to degranulation. To help to understand, tryptase levels must be measured in the blood of the different studied groups. It would be useful to check chymase as well as histamine in the blood sera. Thoses data will demonstrate any degranulation/protease release.
2- The increase of chymase in MC leads to a higher number of chymase+ MCs and chymase+/tryptase+ MCs. Is it due to a phenotype switch of tryptase+ MCs into chymase+/tryptase+ MCs or chymase+ MCs. Furthermore, are those MC newly generated? Please discuss this point.
3-Is the decrease of MC number in general after space flight due to MC death? Migration? Or lost of tryptase/chymase expression? Please discuss this point. It would be useful to stain with another marker/protein expressed by mast cells such as FeRIa or other granule contents.
Minor comments:
1-In materials and methods table 3 please specify the clone of the tryptase antibody used.
Author Response
Рецензент #1
Atiakshin et al. studied the mast cell protease expression in digestive organs of mongolian gerbils after space flight conditions. Using immunohistology staining for tryptase and chymase, they showed that after space flight mast cell number decreased in digestive organs. They observed an increase of chymase expression in MCs (including tryptase+/chymase+ MCs and chymase+ MCs) with a decrease of tryptase expression in MC (tryptase+MCs). I would have some points to address.
The authors thank the referee for the systematic analysis of the article, questions and suggestions. Our answers are attached below.
Major comments:
1- Line 166-169: the authors showed a decrease of tryptase in MCs with some de granulation. It is possible that the tryptase decrease is due to degranulation. To help to understand, tryptase levels must be measured in the blood of the different studied groups. It would be useful to check chymase as well as histamine in the blood sera. Thoses data will demonstrate any degranulation/protease release.
Thanks for the comment. Indeed, this is a very important question, which is highly informative for interpreting the state and functional role of mast cells after a space flight at the system level. It should be noted that experiments on biological objects in space are quite rare. This is due to the complexity of any biological program in orbital flight. The 12-day flight on the spacecraft "Photon-M" No. 3 took place in 2007 and, unfortunately, we did not have any blood samples of Mongolian gerbils that we could subject to the necessary analysis to determine tryptase, chymase and histamine. However, we will definitely consider the reviewer's wishes in future experiments or ground-based models to simulate the physiological effects of weightlessness in ground conditions.
2- The increase of chymase in MC leads to a higher number of chymase+ MCs and chymase+/tryptase+ MCs. Is it due to a phenotype switch of tryptase+ MCs into chymase+/tryptase+ MCs or chymase+ MCs. Furthermore, are those MC newly generated? Please discuss this point.
Thanks for the question. We have included the answer in the "Discussion" section, and the insert is highlighted in red (lines 269-280):
Thus, increase of chymase in MC leads to a higher number of chymase+ MCs and chymase+/tryptase+ MCs. The detected changes in the MC population are most likely due to changes in gene expression in MCs of the gastrointestinal tract. Based on earlier studies, we suggest a high degree of plasticity in the MC protease phenotype. In response to external stimuli, MCs can significantly change the biogenesis of preformed secretome components. Obviously, under space flight conditions, the increased need for the physiological effects of chymase leads to the activation of chymase gene transcription processes in a certain pool of MCs localized in the digestive system, which is manifested by an increase in the Tryptase+Chymase+ phenotype. In addition, an increase in the number of MCs without tryptase expression (with the Tryptase-Chymase+ phenotype) suggests that migrating MC precursors from the bone marrow into the digestive system already have predominant chymase expression.
3-Is the decrease of MC number in general after space flight due to MC death? Migration? Or lost of tryptase/chymase expression? Please discuss this point. It would be useful to stain with another marker/protein expressed by mast cells such as FeRIa or other granule contents.
We thank the referee for an important question to determine the total population of MCs in the digestive system. In earlier studies, we analyzed the effectiveness of histochemical and immunohistochemical reactions in the detection of mast cells, including in Mongolian gerbils. Previously, we obtained results that the most effective method was the immunohistochemical approach (Atiakshin D, Samoilova V, Buchwalow I, Boecker W, Tiemann M. Characterization of mast cell populations using different methods for their identification. Histochem Cell Biol. 2017 Jun;147(6 ):683-694 doi: 10.1007/s00418-017-1547-7 Epub 2017 Feb 27. PMID: 28243739). However, at the suggestion of the reviewer, we will carry out the simultaneous detection of Fc Epsilon RI Alpha with other specific proteases and secretome targets with great interest in future studies. Perhaps we will get a method for a more objective determination of the total intraorgan size of the MC population. In addition, we are grateful to the referee for his contribution to the development of the discussion of this article.
Our insert into the discussion (lines 229-263):
“The decrease in the total amount of MC in the examined organs of the digestive system of Mongolian gerbils after space flight may be associated with several factors, including apoptosis, migration, and increased secretion of specific proteases. In particular, in an experiment with ground-based modeling of space flight factors using a rotary cell culture system, it was shown that microgravity negatively regulates the survival and effector function of mast cells (Kim M, 2022). However, the depletion of intracellular reserves of specific proteases should also be considered, which can make it difficult to detect MC by immunohistochemical staining.”
Kim M, Jang G, Kim KS, Shin J. Detrimental effects of simulated microgravity on mast cell homeostasis and function. Front Immunol. 2022 Dec 16;13:1055531. doi: 10.3389/fimmu.2022.1055531. PMID: 36591304; PMCID: PMC9800517.
Minor comments:
1-In materials and methods table 3 please specify the clone of the tryptase antibody used.
Thank you. We have made it.
Reviewer 2 Report
The manuscript presents an experiment on how gastrointestinal MCs function after being exposed to spaceflight and low gravity circumstances with particular focus on tryptase and chymase.
The article is novel, very well written, well readable and scientific research is of high significance
Major issues: none detected
Minor comments:
Line 322 "The decapitation of the animals from the flight experiment group was carried out 21 hours after the landing" I would consider changing to something like " The animals were sacrificed 21 hours after landing"
Refering to Tables and study design: What is actual difference between groups SE and AnOrtSusp?
Author Response
#Рецензент 2
Comments and Suggestions for Authors
The manuscript presents an experiment on how gastrointestinal MCs function after being exposed to spaceflight and low gravity circumstances with particular focus on tryptase and chymase.
The article is novel, very well written, well readable and scientific research is of high significance
The authors thank the referee for the high evaluation of the work performed.
1.Major issues: none detected
2.Minor comments:
2.1.Line 322 "The decapitation of the animals from the flight experiment group was carried out 21 hours after the landing" I would consider changing to something like " The animals were sacrificed 21 hours after landing"
Fixed (now line 345)
2.2. Refering to Tables and study design: What is actual difference between groups SE and AnOrtSusp?
The synchronous experiment was carried out in a model of flight equipment identical to that installed on board the spacecraft "Photon-M" No. 3. At the same time, the synchronous experiment began 2 days after the launch of the Foton-M spacecraft No. 3 and repeated all the conditions that accompanied Mongolian gerbils in space (except for weightlessness and cosmic radiation), namely: light regime, animal feeding regime, temperature regime, noise effects during spacecraft launch and landing, humidity conditions, etc.
Anti-Orthostatic Suspension is a well-known laboratory rodent ground-based model of simulated microgravity with hindlimb unloading during the experiment time. In this case, a long-term change in body position is achieved, which is often used in ground-based experiments in space physiology and medicine.
Round 2
Reviewer 1 Report
The authors did not add any experimental data to respond the questions 1 and 2. However considering that experiments on biological objects in space are quite rare. The authors tried to answer and comment/discuss the data satisfactorily.